# The Circadian Clock, Nutritional Signals and Reproduction: A Close Relationship

**DOI:** 10.3390/ijms24021545

**Published:** 2023-01-12

**Authors:** Masanori Ono, Hitoshi Ando, Takiko Daikoku, Tomoko Fujiwara, Michihiro Mieda, Yasunari Mizumoto, Takashi Iizuka, Kyosuke Kagami, Takashi Hosono, Satoshi Nomura, Natsumi Toyoda, Naomi Sekizuka-Kagami, Yoshiko Maida, Naoaki Kuji, Hirotaka Nishi, Hiroshi Fujiwara

**Affiliations:** 1Department of Obstetrics and Gynecology, Tokyo Medical University, Tokyo 160-0023, Japan; 2Department of Obstetrics and Gynecology, Graduate School of Medical Sciences, Kanazawa University, Kanazawa 920-8641, Japan; 3Department of Cellular and Molecular Function Analysis, Graduate School of Medical Sciences, Kanazawa University, Kanazawa 920-8641, Japan; 4Institute for Experimental Animals, Advanced Science Research Center, Graduate School of Medical Sciences, Kanazawa University, Kanazawa 920-8641, Japan; 5Department of Social Work and Life Design, Kyoto Notre Dame University, Kyoto 606-0848, Japan; 6Department of Integrative Neurophysiology, Graduate School of Medical Sciences, Kanazawa University, Kanazawa 920-8641, Japan; 7Department of Nursing, College of Medical, Pharmaceutical, and Health Sciences, Kanazawa University, Kanazawa 920-8641, Japan

**Keywords:** BMAL1, circadian rhythm, clock gene, miscarriage, pregnancy, reproduction

## Abstract

The circadian rhythm, which is necessary for reproduction, is controlled by clock genes. In the mouse uterus, the oscillation of the circadian clock gene has been observed. The transcription of the core clock gene period (*Per*) and cryptochrome (*Cry*) is activated by the heterodimer of the transcription factor circadian locomotor output cycles kaput (*Clock*) and brain and muscle Arnt-like protein-1 (*Bmal1*). By binding to E-box sequences in the promoters of *Per1/2* and *Cry1/2* genes, the CLOCK-BMAL1 heterodimer promotes the transcription of these genes. Per1/2 and Cry1/2 form a complex with the Clock/Bmal1 heterodimer and inactivate its transcriptional activities. Endometrial BMAL1 expression levels are lower in human recurrent-miscarriage sufferers. Additionally, it was shown that the presence of BMAL1-depleted decidual cells prevents trophoblast invasion, highlighting the importance of the endometrial clock throughout pregnancy. It is widely known that hormone synthesis is disturbed and sterility develops in Bmal1-deficient mice. Recently, we discovered that animals with uterus-specific Bmal1 loss also had poor placental development, and these mice also had intrauterine fetal death. Furthermore, it was shown that time-restricted feeding controlled the uterine clock’s circadian rhythm. The uterine clock system may be a possibility for pregnancy complications, according to these results. We summarize the most recent research on the close connection between the circadian clock and reproduction in this review.

## 1. Introduction

In all mammalian species, including humans, circadian timing is essential for successful female reproduction. For instance, women who have irregular sleep or work schedules have lower fertility and higher chances of miscarriage [1]. Similar to humans, rats exhibit substantial anomalies in ovulation, fertility, and sexual drive as a result of changes to circadian rhythm [2,3]. Infertility/reproductive disorders are one of the main physiological features of animals with clock mutations. Arnt-like protein-1 (*Bmal1*) knockout in arginine vasopressin (AVP) neurons, kisspeptin neurons, GnRH neurons, or the whole body disrupts the timing and pattern of LH secretion, suggesting that the circadian clock system may integrate the HPG axis [4,5]. Circadian rhythms have been reported for the HPG axis in mice, rats, and humans [3,6,7,8]. In this context, clock genes are also expressed within the reproductive organs [9,10], and the rhythmic expression of *Bmal1*, circadian locomotor output cycles kaput (*Clock*), period (*Per*), and cryptochrome (*Cry*) within the uterus during pregnancy has been reported [11]. Moreover, mutations that alter the clock function can cause infertility in female mice [3,7,8]. Previous studies demonstrated that clock genes play important roles in regulating fertility [12,13,14], and those endocrine factors affect these clock genes [15,16,17]. Compared with the large amount of rodent data available, there have, however, been few studies on the relationship between fertility and clock disturbances in humans. Circadian clock dysfunction causes abnormalities in sleep, appetite, and emotional control [18,19,20,21,22]. Similarly, disturbances in circadian rhythms due to jet lag and night shift work are related to an increased frequency in menstrual cycle abnormalities, altered serum gonadotropin levels, and decreased fertility [23,24]. Meta-analyses have revealed associations between night shift work and an increased frequency of miscarriages [25,26]. Similar findings were noted in a study examining the miscarriage rate of pregnant flight attendants who worked during overnight hours [27]. Moreover, long-time workers whose schedules include night shifts during pregnancy have an increased risk of preterm delivery and low birth weight [28,29]. Given this, pregnant workers are no longer required to work at night when medically indicated in Europe and Japan.

The CLOCK-BMAL1 heterodimer induces transcription of Per1/2 and Cry1/2 genes by binding to E-box (CACGTG/T) regions in their promoters. Together with the Clock/Bmal1 heterodimer, Per1/2 and Cry1/2 form a complex that inhibits the transcriptional activity of CLOCK-BMAL1 [30,31]. Significant alterations in the circadian behavioral rhythms have been seen in *Bmal1*-knockout mice, Clock mutant mice, and *Per*- and *Cry*-deficient animals [32,33,34,35]. The suprachiasmatic nucleus (SCN) integrates information from the external light–dark cycle of the sun to entrain the cellular clocks of organs with the external environment [36,37,38]. The SCN is divided into two major parts: the core and the shell SCN. The core SCN contains the cell bodies of vasoactive intestinal polypeptide (VIP) neurons and the shell SCN contains the cell bodies of arginine vasopressin (AVP) neurons [39]. VIP neurons input onto gonadotropin-releasing hormone (GnRH) neurons in the preoptic area (POA), and AVP neurons input onto kisspeptin neurons in the anterior ventral periventricular (AVPV) nucleus [40]. It has been noted that the circadian rhythms of AVPV Kiss1 expression in Kiss1 neurons peaked coincident with LH, suggesting the interactions between the SCN and the reproductive neurons in the female hypothalamic–pituitary–gonad (HPG) axis [41]. The primary subject of this review is how circadian clock disorders cause these abnormalities in reproduction.

## 2. Effects of the Circadian Clock on the Hypothalamic–Pituitary–Gonadal (HPG) Axis and Reproduction

Female reproduction is under circadian control and temporal information is relayed through the HPG axis [13,42,43,44]. The 24 h rotation of the Earth produces regular patterns of environmental modifications, consisting of adjustments in light–dark, changes in temperature, risks of predation, and food availability [45]. The impact of molecular clocks on the HPG axis in relation to female reproduction is well known (Figure 1). The SCN regulates the circadian rhythm of *Kiss1* expression in the AVPV [46]. Importantly, *Bmal1* and other clock genes have also been identified in kisspeptin neurons [47].

### 2.1. Circadian Clock Regulation of the HPG Axis

Kisspeptin regulates the secretion of sex steroids such as estrogen through the HPG axis [48]. Kisspeptin signaling is necessary for the timing of reproductive activity, including the pulsatile and estrous cycle of GnRH [40,49]. It is known that GnRH secretion is regulated by circulating hormones, kisspeptin, and neurotransmitters [50,51,52,53,54]. Daily changes in GnRH cell responsiveness to kisspeptin have also been reported [55]. The sensitivity of the GnRH system to kisspeptin stimulation fluctuates significantly during the day, peaking in the afternoon [56]. Kisspeptin neurons are found in the AVPV and ARC nuclei of the hypothalamus, and the SCN controls the AVPV nucleus. AVPV kisspeptin neurons, whose activity is regulated by SCN signals in an E_2_-dependent manner, are responsible for controlling LH surge [41]. Although kisspeptin neurons in the ARC are more influenced by E_2_ and leptin than SCN signals [57,58], the effects of circadian dysregulation (e.g., skipping breakfast, shift work, and transmeridian travel) as a factor affecting infertility cannot be overlooked [59,60,61] (Figure 2). For the GPR54 receptor, kisspeptin serves as the endogenous agonist. It was determined that GPR54 expression can become rhythmic when E2 levels are raised, a behavior that appears to be controlled by intracellular ERβ receptors [62].

### 2.2. Ovarian Circadian Clock

In the rat ovary, clock genes associated with the ovulation cycle have been identified. The day of proestrus sees a considerable increase in BMAL1 expression following the LH surge [15]. In follicular development, *Per1* and *Per2* mRNA are localized to steroidogenic cells in preantral, antral, and preovulatory follicles, corpora lutea, and interstitial glandular tissue by in situ hybridization histochemistry [63]. Furthermore, *Per1* and *Per2* mRNA and proteins oscillate in a circadian manner in follicles, granulosa cells, and theca cells. In contrast, LH promotes *Per1* as well as *Bmal1* expression in the ovary [15]. These clock genes display different amplitudes at different stages of the estrus cycle, suggesting endocrine control of the circadian clock [16]. Importantly, gonadotropins also control the ovarian clock, which is supported by experiments indicating that the administration of gonadotropins can synchronize isolated ovaries [64]. Using a *Per1*-luciferase reporter assay, circadian rhythms were noted in the ovaries, and clock gene phasing was observed in response to LH and FSH [64]. In addition, the ovaries of a mouse model of polycystic ovarian syndrome (PCOS) were shown to have an anomaly for the time of *Per2* rhythm [65].

### 2.3. Endometrial Circadian Clock

An analysis of the relationship between the decidual circadian rhythm and recurrent miscarriage showed that *BMAL1* expression in the human decidua during early pregnancy was decreased in patients that experienced recurrent miscarriage [66]. In particular, knockdown may impair the regulation of trophoblast invasion by decidual cells, disrupting proper placenta formation. Polymorphisms in the circadian clock genes are also associated with a higher risk of miscarriage, and gene variants were found in *BMAL1* and *NPAS2* [67]. Furthermore, progesterone is known to affect the peripheral endometrial clock rhythm in humans. When progesterone acts on the endometrium and decidualization occurs, the level of *PER1* in the endometrium increases [68]. The microenvironment of the uterus responds to circadian rhythms and adapts to physiological functions. During pregnancy, the fetus is continuously exposed to hormonal and nutritional signals in the maternal endometrium [69,70]. These results suggest that the circadian rhythms play significant roles in reproduction.

### 2.4. Animal Studies on the Circadian Clock and Reproductive Function

Consistent with its expression pattern, global *Bmal1* knockout mice were found to have significantly reduced ovulation compared with control mice [8]. Global *Bmal1* knockout mice were also known to be infertile [5,8,71] (Table 1). Global *Bmal1* knockout mice also showed delayed puberty and abnormal estrous cycles [5,72], and the deletion of *Bmal1* was shown to reduce progesterone levels [5,73]. Later, the failure of embryo implantation in steroidogenic factor-1 (SF-1) expression-dependent *Bmal1*-deleted female mice (*Bmal1^SF1d/d^*) was shown to be rescued by P_4_ supplementation or normal ovarian transplantation, demonstrating that insufficient ovarian P_4_ production is one of the primary causes of infertility in *Bmal1* knockout female mice [73]. Additional studies were carried out involving the conditional knockout of *Bmal1* in ovarian granulosa cells or theca cells [7] (Table 1). Theca cells are the pacemakers that regulate ovulation timing and transient sensitivity to LH [14]. In these conditional knockout mice, transient susceptibility to LH was found in littermate controls and granulosa cell-specific *Bmal1* knockout mice, but not in theca cell-specific *Bmal1* knockouts [7,74]. This indicated that follicle development and ovulation are affected by circadian rhythm disfunction in theca cells (Table 1).

Recently, we generated mice with a conditional deletion (cKO) of uterine *Bmal1* to examine the pathogenic functions of the uterine clock genes during pregnancy [75]. We found that cKO mice could achieve embryo implantation but could not maintain pregnancy. A histological analysis of their placentas showed that the maternal vascular spaces failed to form properly. In contrast to WT mice, cKO mice expressed scarce levels of the immunosuppressive NK marker CD161 in the spongiotrophoblast layer where maternal uNK cells are in close contact with the fetal trophoblast. These data suggest that *Bmal1* plays a significant role in the reproductive organs (Table 1).

**Table 1 ijms-24-01545-t001:** Distinct reproductive characteristics of *Bmal1* mutant mice.

Mutant Mice	Phenotypes of Reproduction	References
**Conventional *Bmal1* KO**	Delayed puberty; females have longer estrous cycles; infertile	[5,76]
**Gonadotrope *Bmal1* KO**	Irregular estrous cycle; fertile	[7,77]
**Granulosa cell *Bmal1* KO** **(GCKO)**	Normal ovarian morphology and a typical estrous cycle; fertile	[7]
**Ovarian steroidogenic cells *Bmal1* KO**	Typical puberty; early pregnancy loss; infertile	[73]
**Theca cell *Bmal1* KO**	Fewer offsprings and increased mating failure; regular estrous cycle; subfertile	[7]
**Uterine *Bmal1* KO**	Reducing placental vascularization and causing fetal mortality within the uterus; subfertile	[75]

Similarly, *Per1* and *Per2* knockout mice experienced reduced reproductive rates because of estrous cycle irregularities [78,79]. Moreover, in *Per1*-*Per2* double knockout mice, the follicular reserve was depleted, resulting in infertility [80]. Mice with a dominant negative mutation in *Clock* (*Clock* Δ19/Δ19 mice) were generated to investigate the molecular mechanisms governing circadian clocks [81]. These mice are capable of producing the BMAL1-CLOCK dimer, but possess a defective form of the CLOCK protein that is unable to regulate *Per* and *Cry* expression, resulting in the loss of the feedback loop for circadian clock genes [81]. *Clock* Δ19/Δ19 mice are also overweight and develop symptoms of metabolic syndrome under high-fat diet (HFD) conditions [82]. This obesity-induced phenotype is associated with feeding during rest time. Untimely feeding is associated with obesity and excess body weight in mice and humans [83,84]. Fasting is also involved in circadian rhythm accommodation or dysregulation. Time-restricted feeding (TRF) in which food access is restricted to the dark phase has been reported to protect mice from obesity, fatty liver, hyperinsulinemia, and inflammation when they are fed an HFD [85,86,87]. Rodents fed an HFD ad libitum showed changes in circadian rhythms compared with rodents fed an HFD with TRF [85,88]. This suggests that feeding affects the circadian clock. In addition to the loss of a circadian rhythm, these mice were also reported to have increased risks of stillbirth and neonatal death compared with controls [89].

The pars tuberalis is situated between the anterior lobe of the pituitary gland and the median eminence. It has been demonstrated that melatonin acts as a photoperiodic signal, synchronizing an endogenous oscillator in the pars tuberalis to the photoperiod [90]. Thyroid-stimulating hormone beta (TSH) cells are found in the pars tuberalis, which also trigger the secretion of TSH. TSH promotes triiodothyronine synthesis, which helps gonadotropin-releasing hormone-I release, luteinizing hormone and follicle, stimulating hormone release [91]. Recent research has shown that pars tuberalis controls seasonal reproduction with its TSH secretion [92,93].

In diurnal primates, labor is often initiated at night, consistent with the increased sensitivity to oxytocin that causes pregnancy-related uterine contractions [94,95]. This suggests that circadian rhythms alter uterine sensitivity to oxytocin [96]. Furthermore, studies in rodents have shown that the uterus has a functional peripheral circadian clock [17,97,98]. It has also been suggested that embryo implantation and delivery are controlled by a peripheral circadian clock in the uterus [99,100]. Maternal myometrium and the bladder-specific deletion of *Bmal1* cause the mistiming of labor onset [101]. While control mice gave birth early in the morning [29], maternal myometrium- and bladder-specific *Bmal1* knockout mice had 28% more daytime births than control mice, demonstrating that the peripheral circadian clock is involved in the timing of labor [29]. These data suggest the importance of circadian clocks in reproduction.

## 3. The Circadian Clock System as a Link between Nutritional Signals and Reproduction

Reproduction is critical for species survival. Nevertheless, under certain environmental conditions, reproductive activity is suppressed. Many organisms, together with humans, adaptively reduce reproductive activity during periods of starvation and/or anorexia [102,103]. Inadequate dietary restrictions are known to adversely affect the rhythmic secretion of luteinizing hormone (LH) [4], ovarian development [5], and decreased human gonadotropin levels [104,105,106]. Food restriction inhibits both GnRH pulse activity and gonadotropin secretion, resulting in insufficient gonadotropin for folliculogenesis [107,108]. This ultimately results in delayed puberty and the suppression of ovulation when the food supply is insufficient [109].

Feeding rhythms are important for animals because food-entrainable oscillators are located within peripheral tissues, and these peripheral oscillators are independent of the SCN [110,111,112]. We found that time-restricted feeding regulates the circadian rhythm of the uterine clock that is synchronized throughout the uterine body [113]. Furthermore, we postulated that breakfast skipping impairs reproductive function by disrupting the circadian clock [114,115]. In modern society, breakfast skipping is a common habit. Previously, we discovered that skipping breakfast is related to dysmenorrhea [116], and later studies have also revealed a similar correlation between skipping breakfast and dysmenorrhea [117,118,119,120]. Experiments in mice were conducted in which feeding was limited to two meals per day at specified intervals (16 and 8 h). These studies found that the circadian clock was reset by a longer interval (16 h fast) than a shorter interval (8 h fast) between meals [88,121]. In general, breakfast corresponds to the start of one’s daily activities, and skipping breakfast interferes with circadian clocks [116,122,123,124]. This suggests that breakfast has the greatest impact on the chronobiology of the daily diet in humans, and skipping breakfast has been proposed to affect the reproductive system [120,125,126].

## 4. The Circadian Clock and Puberty

Proper timing of sexual maturation is necessary for reproduction [127,128,129]. Circadian regulation of the reproductive organs is associated with the timing of GnRH release and gonadotropin secretion, and these processes affect sexual maturation [77,130]. Moreover, human and animal puberty relies on complex endocrine regulation [131]. In European sea bass, a prolonged photoperiod delays or prevents puberty and the release of the hormones associated with reproduction [132,133]. One variable in female puberty is the age at menarche, and the timing of menarche is impacted by light. In women who are blind with loss of light perception, menarche occurs earlier than in women with normal light perception [134]. In addition, women are more likely to experience precocious puberty than men [135,136]. From a disease perspective, the associations between the timing of puberty and the risk of developing endometrial or breast cancer in women and prostate cancer in men have been described [137]. Thus, focusing on circadian rhythms may provide clues to preventing and/or treating these diseases.

Other factors affecting sexual maturation are endocrine-disrupting chemicals (EDCs). EDCs are substances that can mimic hormones in the body and are found in common household products. EDCs bind to hormone receptors and cause activation or suppression of natural hormones or alter the breakdown of natural hormones, thereby causing changes in normal hormonal signaling. Puberty is a complex developmental stage in which physical changes promote sexual maturation, and this process is sensitive to hormonal disruptions. EDCs have been reported to be involved in pubertal-onset variability [138] and can enter the body through drinking, eating, breathing, or direct contact [139]. Exposure to EDCs with estrogenic and/or anti-androgenic effects can disrupt the reproductive tract and sexual maturation [140]. Over the last 200 years, the timing of pubertal onset has changed. The age of menarche has been reduced from 17 in the early 19th century to 13 in the 1950s [141]. The liver of adult male Wistar rats treated with 4-hydroxy-2,3,3’,4’,5-pentachlorobiphenyl showed altered expression of the clock genes including BMAL1 [142]. Moreover, various studies have demonstrated changes in circadian clock-gene expression and the endocrine system after exposure to EDCs, and the importance of this is now clear [143,144,145].

## 5. Conclusions

In conclusion, elucidating the factors that modify circadian clocks in reproductive organs will provide clues to treating reproductive dysfunction. Moreover, it may suggest strategies for optimizing existing therapeutic interventions. We expect that the appropriate re-establishment of the networks governing circadian rhythms and the reproductive cycle in early life will help prevent future obstetric and gynecological diseases. The influence of circadian rhythms governing protein translation on the regenerative capacity of tissues must be considered in future studies of regeneration.

## Figures and Tables

**Figure 1 ijms-24-01545-f001:**
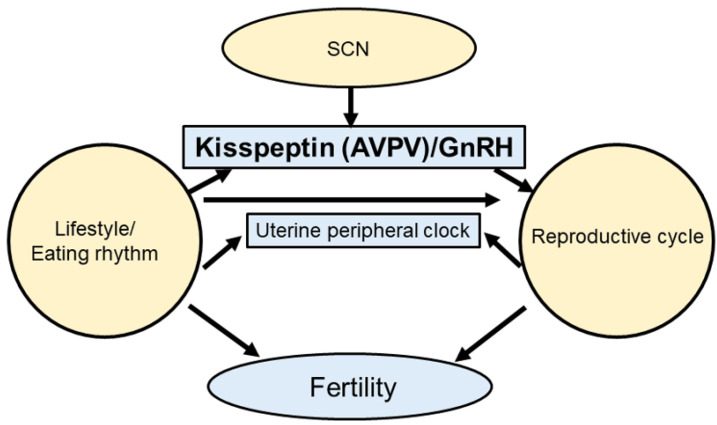
Synchrony of the circadian clock and reproduction. The central clock controls kisspeptin secretion. Feeding rhythms also control hormone secretion. Moreover, circadian clocks and reproductive cycles affect fertility.

**Figure 2 ijms-24-01545-f002:**
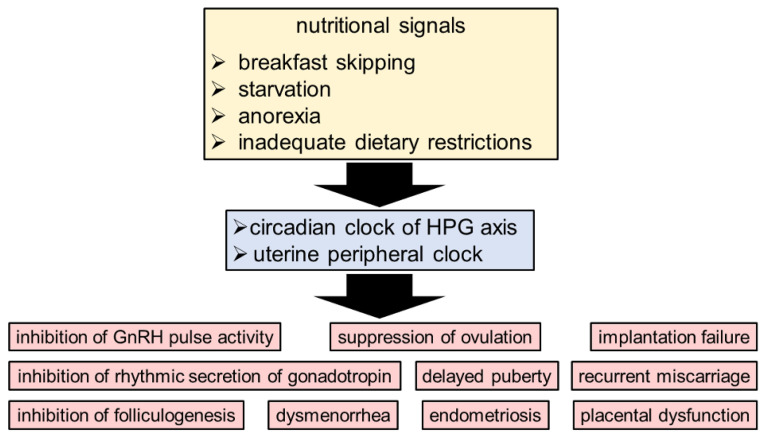
The circadian clock system as a relationship between reproductive health and nutritional signals. The circadian rhythm in the HPG axis and uterus is negatively impacted by nutritional signals such as skipping breakfast, starvation, anorexia, and inadequate dietary restrictions.

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
