# Peer review of "The Circadian Clock, Nutritional Signals and Reproduction: A Close Relationship"

_ijms, 2023, doi:10.3390/ijms24021545_

Round 1
Reviewer 1 Report (Previous Reviewer 2)
This is an article I have commented on in November.The circadian rhythm, which is necessary for mammalian or avian reproduction, is the medium for environmental and animal reproductive activities. This manuscript reviewed the regulation of circadian rhythms and nutritional signals on key reproductive activities. The article has made some modifications and improvements on the basis of previous comments, I have no comments except English writing now. I think it has reached the requirement for publication on the IJMS.
Author Response
Manuscript #IJMS-2129379R1
Title: The Circadian Clock, Nutritional Signals and Reproduction: A Close Relationship
Reviewer’s Comments to Author:
Reviewer: 1
This is an article I have commented on in November. The circadian rhythm, which is necessary for mammalian or avian reproduction, is the medium for environmental and animal reproductive activities. This manuscript reviewed the regulation of circadian rhythms and nutritional signals on key reproductive activities. The article has made some modifications and improvements on the basis of previous comments, I have no comments except English writing now. I think it has reached the requirement for publication on the IJMS.
Thank you. We utilized the English Pre-Editing service in IJMS (https://www.mdpi.com/authors/english) in accordance with your advice to improve our manuscript. We greatly appreciate the opportunity to improve the quality of our manuscript and would like to thank you for your insightful comments.
Sincerely,
Masanori Ono MD, PhD.
Department of Obstetrics and Gynecology, Tokyo Medical University
6-7-1, Nishishinjuku, Shinjuku, Tokyo 160-0023, Japan.
Tel: 81-3-3342-6111
Fax: 81-3-3348-5918
masanori@tokyo-med.ac.jp

Reviewer 2 Report (New Reviewer)
Dear authors!
The analysed problem is actual and is important for practical medicine.
For polish the article, I suggest you to improve the Figure 2. Its title is general, but it reflects only effect of uterine peripheral clock. However, in the text you analysed separately the ovarian circadian clock and endometrium one. “Abnormalities in circadian clock” – which one? It should be added schematically the reason of these disorders – change in genes expression or losing of genes, or changes in diet, nutritional regime etc.
The weakness of your review is limitation of analyze of human studies. I would like to suggest you to continue your investigations and collect more data related to human studies and difference between male and female circadian clock.
Author Response
Manuscript #IJMS-2129379R1
Title: The Circadian Clock, Nutritional Signals and Reproduction: A Close Relationship
Reviewer’s Comments to Author:
Reviewer: 2
The analyzed problem is actual and is important for practical medicine.
We sincerely appreciate this opportunity to improve our manuscript’s quality and would like to thank you for the constructive comments on this work.
For polish the article, I suggest you to improve the Figure 2. Its title is general, but it reflects only effect of uterine peripheral clock. However, in the text you analyzed separately the ovarian circadian clock and endometrium one. “Abnormalities in circadian clock” – which one? It should be added schematically the reason of these disorders – change in genes expression or losing of genes, or changes in diet, nutritional regime etc.
Thank you. Figure 2 was changed to the new one, "The circadian clock system as a relationship between reproductive health and nutritional signals." Then, the insertion position of Figure 2 was set to line 233, and as you advised, we improved it to make it easier to understand.
The weakness of your review is limitation of analyze of human studies. I would like to suggest you to continue your investigations and collect more data related to human studies and difference between male and female circadian clock.
Thank you for your valuable suggestions. We would like to proceed with human investigations, collect more data related to human studies, and also plan to study the difference between male and female circadian clock. We recognize that these are very important themes.
Sincerely,
Masanori Ono MD, PhD.
Department of Obstetrics and Gynecology, Tokyo Medical University
6-7-1, Nishishinjuku, Shinjuku, Tokyo 160-0023, Japan.
Tel: 81-3-3342-6111
Fax: 81-3-3348-5918
masanori@tokyo-med.ac.jp

This manuscript is a resubmission of an earlier submission. The following is a list of the peer review reports and author responses from that submission.
Round 1
Reviewer 1 Report
Re: Manuscript #ijms-2006312, Title: The circadian clock and reproduction: a close relationship
Summary
The main theme of this review manuscript seems to be circadian clock roles in reproduction. The main parts are composed by 3 parts, which are section 2, 3, and 4. The section 2 addresses circadian clock system in HPG axis and reproduction. The section 3 introduces the correlation between skipping breakfast and reproduction with linking those by circadian rhythmicity. And the section 4 introduces the roles of circadian clocks in sexual maturation. Although the section 2 is commonly seen everywhere as review papers, the section 3 and 4 provides originality in this manuscript.
However, there are so many issues to support their main theme of this manuscript and it is quite necessary to address the concerns below.
Major comments
Overall
I suggest the authors to rewrite some parts in more descriptive, explainable strategy. Especially the subsection 2-1, 2-2, and 2-3, each sentence is fragmentary, and each subsection is too short. In many parts, the authors refer papers as previous observation, without any introductive descriptions of those works. It makes very hard to understand what the authors’ message without visiting cited papers. Especially, since this is review paper, not research articles, it requires to explain those cited works. Please do not just say “previously observed”, “previously reported”, or else, with expecting the readers to visit those papers to understand what you want to say. At lease, minimum, but explainable sentences are required. Otherwise, to understand this manuscript, readers need to check cited papers and return to this manuscript.
Title: What is the main theme of this review paper? After reading throughout current manuscript, I don’t feel any match of this title and the contents in the main body. Also, how close between circadian clock system and reproduction in this specific review paper? There are numerous papers/review papers addressing the roles of the circadian clock system in reproduction. This title is too vague. It would be better to reconsider and have more specific words in the title.
Abstract: The message authors want to emphasize is vague. Here is one of the examples.
Line 21: What exactly means “circadian rhythm of the mouse uterus.”? What kind of “rhythm” referring authors to?
One of the reasons I can assume is the way to write. Each sentence is cut off in short sentence and no relevancy between. For example, line 20-22, second sentence indicates mouse information. Then authors suddenly jump onto human information in the next sentence. What is the linkage between those two sentences?
There are more points need to improve throughout the abstract section. I suggest authors to revise the whole abstract thoroughly.
Section 1. Introduction
Overall, the authors want to re-compose whole section of introduction to emphasize the reproduction system and the influences of circadian rhythms (e.g., shift work, jet lug, travelling over the time zone) first. Then introduce the circadian clock system as one of the key mechanisms to maintain reproductive physiology and functions. The initiation of the sentences with the circadian clock system and clock molecule introductions can be seen everywhere. To be originated, I strongly recommend authors to put emphasis on the reproduction system and its concerns under circadian misalignment.
Here is one of the example: begin with reproduction system and its concerns and abnormalities > physiological concern and cause of reproduction (sterility, miscarriage, etc.) > molecular mechanisms, including circadian clock system > some evidence of linkage between reproduction and circadian clock system (reproductive concerns of shift workers, nurses, and so on).
I believe the review paper published from the labs of Alexander S. Kauffman, Rae Silver, or Lance J Kriegsfeld are the great examples.
Section 2
Overall, the information in subsections 2-1, 2-2, and 2-3 are insufficient to conclude each theme. It is necessary to enrich with a more descriptive and strategic way of writing.
Line 82-85: How come the authors can conclude as such with those pieces of independent, unlinked information?
Line 84-85: Review is not the place to give the authors’ assumption. In fact, there are numerous evidence to develop infertility in model mice with circadian clock molecules knockout, circadian disruption, and sleep deprivation.
2-1
The title “Circadian clock regulation of gonadotropin” and contents doesn’t quite match. This title may give the impression of circadian clock directly regulate gonadotropin production and secretion. I suggest authors to replace the word of “gonadotropin” with “HPG axis” or else. Also, the information about the circadian clock system in Kisspeptin neurons, GnRH neurons, or HPG axis are too small. It is required to be enriched. I also recommend adding GPR54.
2-3
Line 120-123: First sentence sounds like suggesting Bmal1 downregulation may be involved in recurrent miscarriage (no citation). However, second sentence sounds like suppression of Bmal1 may help to attenuate decidualization. It sounds inconsistent for me.
Line 132: It is insufficient, or hard to understand, to conclude as the authors mentioned.
2-4
This section seems to be well written in a descriptive manner.
If the authors compose animal model studies as one section here, I suggest the authors creating one section to focus on human case above. The subsections of 2-1~2-3 are too small and containing animal model studies which mostly overlapped with this section. Reorganize the subsection 2-1~2-3 as a review of human case would be appreciated.
Section 3
In this section, especially the second paragraph, it is suggestive, but not conclusive information to explain the correlation between skipping breakfast and the circadian clock system. If this is the review for “the correlation between nutrition and reproduction”, this is well written. However, the authors put emphasis on the circadian clock system as a link between nutrition and reproduction, the information written in this section seems insufficient. What exactly occur when breakfast is skipped in the circadian clock system and reproduction system (other than dysmenorrhea)? What is the core of the connection? This paragraph begins with the authors’ hypothesis and ends with a proposal. It would be great if the authors add one or two sentence(s) to conclude this paragraph.
Line 207-214: Here the authors explain the effects of skipping breakfast or longer duration of fasting in circadian clocks. However, nutritional signals mostly influence circadian clock systems in peripheral metabolic organs (e.g., liver, muscle, etc.). Is there any reasonable evidence to link skipping breakfast with reproduction through the circadian clock system? For example, circadian clock expression patterns in ovary or uterus, Kisspeptin circadian expression patterns, etc.
Section 4
The subject of this section is very interesting. However, as same as section 3, it is too weak to link sexual maturation between circadian clocks with current style. Do EDCs influence in circadian oscillation? Is there any EDCs as ligands directly regulate the circadian clock system (e.g., RORs/REV-ERBs ligands)? Is there any evidence to reveal EDCs influence circadian clock system and its downstream reproduction system (e.g., Kisspeptin)?
Section 5
I recommend the authors to reconsider and rewrite this section after revising whole part of this manuscript, since the sections above seem too weak to conclude as current states.
Line 224: I suggest the authors to say more positive way rather than saying “We hypothesize that”. The hypothesis is a hypothesis. “We expect ~/We hope for ~” would sound better.
Minor concerns
Line 35-37: “core of molecular interaction in the circadian rhythm” sounds inaccurate. It is true that the BMAL1/CLOCK and PERs/CRYs interact each other to regulate circadian gene expression patterns as a core mechanism of the circadian clock system. What exactly authors want to explain with the words of “core of molecular interaction”?
Line 38-39: The paper authors referred is the paper for circadian clock system in bone metabolism. Is that paper the most relevant?
Line 39: What is “The central clock in the hypothalamic suprachiasmatic nucleus (SCN)” refers to?
Line 40: What kind of “information” we can receive “from the external light-dark cycle”?
Line 42: “the core SCN and the shell SCN.” > “the core and shell SCN.”
Line 43 and 44: “cellular bodies” > “cell bodies”
Line 47-49: How important? No explanation of that report? How about males?
Line 50-61: This part is one of the cores of this manuscript. I strongly suggest authors to reorganize this part, especially enrich the human part, and bring to the first section of the introduction. Especially, even there are only “few studies” conducted, it is very important to indicate the evidence of reproductive concerns which possibly link to circadian rhythms.
Line 62-71: Again, this part contains some important information related to human physiology and pathology. I recommend authors to marge this section with Line 59-61 and bring to the opening sentence.
Line 94-95: It seems inconsistent with the referred paper.
Line 97-100: Is “dysfunction” the word exactly the authors want to say? “dysregulation”? or “dysfunction caused by ….”? Please check it.
Line 108: “local peripheral circadian regulation” of what? Follicular development itself? Does that event such dynamic?
Line 109: “in a circadian rhythm” > “in a circadian manner”
Line 111-112: “These clock genes have different amplitudes that depend on the estrous cycle” > “These clock genes display (show) different amplitudes (and rhythmic patterns?) at different stages of estrus cycle”
Line 112-114: I suggest the authors to merge line 112-113 and 113-114 as one sentence, remove unnecessary part and, if possible, add a bit more information about the study in Ref#57.
Line 114-116: How abnormal?
Line 116: “Per1 luciferase reporter assay” supposed to be “Per1-luc (or Per1-luciferase) reporter assay”. With or without “-”, the meaning is totally different.
Line 125: What exactly means in “genes that control circadian rhythms”? If the authors want to say, “circadian clock genes” or “core clock genes”, it is a bit circumlocutory.
Line 134-135: It is already mentioned the circadian clock expressions above. Is there any specific reason not to mention here?
Line 144: “conditional knock out” > “conditional knockout”
Lie 173: “Time-limited feeding (TRF)” suppose to be “Time-restricted feeding (TRF)”
There are so many minor concerns in the current version. I will check further when I have a chance to review revised version.
Author Response
Manuscript #ijms-2006312R1
Title: The Circadian Clock, Nutritional Signals and Reproduction: A Close Relationship
Reviewer’s Comments to Author:
Reviewer: 1
Summary
The main theme of this review manuscript seems to be circadian clock roles in reproduction. The main parts are composed by 3 parts, which are section 2, 3, and 4. The section 2 addresses circadian clock system in HPG axis and reproduction. The section 3 introduces the correlation between skipping breakfast and reproduction with linking those by circadian rhythmicity. And the section 4 introduces the roles of circadian clocks in sexual maturation. Although the section 2 is commonly seen everywhere as review papers, the section 3 and 4 provides originality in this manuscript.
We thank the reviewer for the positive comments on our manuscript and greatly appreciate this opportunity to improve our manuscript.
Major comments
I suggest the authors to rewrite some parts in more descriptive, explainable strategy. Especially the subsection 2-1, 2-2, and 2-3, each sentence is fragmentary, and each subsection is too short. In many parts, the authors refer papers as previous observation, without any introductive descriptions of those works. It makes very hard to understand what the authors’ message without visiting cited papers. Especially, since this is review paper, not research articles, it requires to explain those cited works. Please do not just say “previously observed”, “previously reported”, or else, with expecting the readers to visit those papers to understand what you want to say. At lease, minimum, but explainable sentences are required. Otherwise, to understand this manuscript, readers need to check cited papers and return to this manuscript.
Thank you for your comments; We have modified the manuscript based on the comments you provided. To make it easier for readers to understand, we added the contents of the cited references, particularly in subsections 2-1, 2-2, and 2-3.
Title: What is the main theme of this review paper? After reading throughout current manuscript, I don’t feel any match of this title and the contents in the main body. Also, how close between circadian clock system and reproduction in this specific review paper? There are numerous papers/review papers addressing the roles of the circadian clock system in reproduction. This title is too vague. It would be better to reconsider and have more specific words in the title.
Thank you. As the reviewer suggested, we included “nutritional signals” in the title to clarify the originality of our review.
Line 21: What exactly means “circadian rhythm of the mouse uterus.”? What kind of “rhythm” referring authors to? One of the reasons I can assume is the way to write. Each sentence is cut off in short sentence and no relevancy between. For example, line 20-22, second sentence indicates mouse information. Then authors suddenly jump onto human information in the next sentence. What is the linkage between those two sentences?
Thank you. We rewrote abstract as follows. “The circadian rhythm, which is necessary for reproduction, is controlled by clock genes. In the mouse uterus, oscillation of the circadian clock gene has been observed. The transcription of core clock genes period (Per) and cryptochrome (Cry) is activated by the heterodimer of the transcription factors circadian locomotor output cycles kaput (Clock) and brain and muscle arnt-like protein-1 (Bmal1). These Per and Cry proteins interact with Clock/Bmal1 complexes to form heterodimers and suppress Clock and Bmal1 transcription. A 24-hour cycle is produced in the cells by this negative feedback loop, which is typically synchronized within the organ. Endometrial BMAL1 expression levels have been shown to be lower in recurrent miscarriage sufferers in humans. Additionally, it was shown that the presence of BMAL1-depleted decidual cells prevents trophoblast invasion, highlighting the importance of the endometrial clock throughout pregnancy. It is widely known that hormone synthesis is disturbed, and sterility develops in Bmal1-deficient mice. Recently, we discovered that animals with uterus specific Bmal1 loss also had poor placental development, and these mice also had intrauterine fetal death. Furthermore, it was shown that time-restricted feeding controlled the uterine clock's circadian rhythm. The uterine clock system may be a possibility for pregnancy complications, according to these results. We give a summary of the most recent research on the close connection between the circadian clock and reproduction in this review.”
Section 1. Introduction
Overall, the authors want to re-compose whole section of introduction to emphasize the reproduction system and the influences of circadian rhythms (e.g., shift work, jet lug, travelling over the time zone) first. Then introduce the circadian clock system as one of the key mechanisms to maintain reproductive physiology and functions. The initiation of the sentences with the circadian clock system and clock molecule introductions can be seen everywhere. To be originated, I strongly recommend authors to put emphasis on the reproduction system and its concerns under circadian misalignment. Here is one of the examples: begin with reproduction system and its concerns and abnormalities > physiological concern and cause of reproduction (sterility, miscarriage, etc.) > molecular mechanisms, including circadian clock system > some evidence of linkage between reproduction and circadian clock system (reproductive concerns of shift workers, nurses, and so on). I believe the review paper published from the labs of Alexander S. Kauffman, Rae Silver, or Lance J Kriegsfeld are the great examples.
Thank you. We reorganized Introduction as follows. “In all mammalian species, including humans, circadian timing is essential for successful female reproduction. For instance, women who have irregular sleep or work schedules had lower fertility and higher chances of miscarriage [1]. Similarly, to humans, rats exhibit substantial anomalies in ovulation, fertility, and sexual drive as a result of changes to circadian rhythm [2, 3]. Infertility/reproductive disorders are one of the main physiological features of animals with clock mutations. Arnt-like protein-1 (Bmal1) knockout in arginine vasopressin (AVP) neurons, kisspeptin neurons, GnRH neurons, or the whole body disrupts the timing and pattern of LH secretion, suggesting that the circadian clock system may integrate the HPG axis [4, 5]. Circadian rhythms have been reported for the HPG axis in mice, rats, and humans [3, 6-8]. In this context, clock genes are also expressed within reproductive organs [9, 10] and the rhythmic expression of brain and muscle arnt-like protein-1 (Bmal1), circadian locomotor output cycles kaput (Clock), period (Per), and cryptochrome (Cry) within the uterus during pregnancy has been reported [11]. Moreover, mutations that alter clock function can cause infertility in female mice [3, 7, 8]. Previous studies demonstrated that clock genes play important roles in regulating fertility [12-14] and that endocrine factors affect these clock genes [15-17]. Compared with the large amount of rodent data available, however, there have been few studies on the relationship between fertility and clock disturbances in humans. Circadian clock dysfunction causes abnormalities in sleep, appetite, and emotional control [18-22]. Similarly, disturbances in circadian rhythms due to jet lag and night shift work are related to an increased frequency in menstrual cycle abnormalities, altered serum gonadotropin levels, and decreased fertility [23, 24]. Meta-analyses have revealed associations between night shift work and an increased frequency of miscarriages [25, 26]. Similar findings were noted in a study examining the miscarriage rate of pregnant flight attendants who worked during overnight hours [27]. Moreover, long-time workers whose schedules include night shifts during pregnancy have an increased risk of preterm delivery and low birth weight [28, 29]. Given this, pregnant workers are no longer required to work at night when medically indicated in Europe and Japan.
The transcription-translation feedback loop involving BMAL1/CLOCK and PER/CRY interact each other to regulate circadian gene expression patterns [30, 31]. Significant alterations in the circadian behavioral rhythms have been seen in Bmal1-knockout mice, Clock mutant mice, and Per- and Cry-deficient animals [32-35]. The suprachiasmatic nucleus (SCN) integrates information from the external light-dark cycle of the sun to entrain the cellular clocks of organs with the external environment [36-38]. The SCN is divided into two major parts: the core and the shell SCN. The core SCN contains the cell bodies of vasoactive intestinal polypeptide (VIP) neurons and the shell SCN contains the cell bodies of arginine vasopressin (AVP) neurons [39]. VIP neurons input onto gonadotropin-releasing hormone (GnRH) neurons in the preoptic area (POA), and AVP neurons input onto kisspeptin neurons in the anterior ventral periventricular (AVPV) nucleus [40]. It has been noted that the circadian rhythms of AVPV Kiss1 expression in Kiss1 neurons peaked coincident with LH, suggesting the interactions between the SCN and reproductive neurons in the female hypothalamic-pituitary-gonad (HPG) axis [41]. The primary subject of this review focuses on how circadian clock disorders cause these abnormalities in reproduction.”
Section 2
Overall, the information in subsections 2-1, 2-2, and 2-3 are insufficient to conclude each theme. It is necessary to enrich with a more descriptive and strategic way of writing.
We included the contents of the cited references to conclude each theme as follows.
“2.1. Circadian clock regulation of HPG axis
Kisspeptin regulates the secretion of sex steroids, such as estrogen, through the HPG axis [42]. Kisspeptin signaling is necessary for the timing of reproductive activity, including the pulsatile and estrous cycle of GnRH [40, 43]. It is known that GnRH secretion is regulated by circulating hormones, kisspeptin, and neurotransmitters [44-48]. Daily changes in GnRH cell responsiveness to kisspeptin have also been reported [49]. The sensitivity of the GnRH system to kisspeptin stimulation fluctuates significantly during the day, peaking in the afternoon [50]. Kisspeptin neurons are found in the AVPV and ARC nuclei of the hypothalamus, and the SCN controls the AVPV nucleus. AVPV kisspeptin neurons, whose activity is regulated by SCN signals in an E2-dependent manner, are responsible for controlling LH surge [41]. Although kisspeptin neurons in the ARC are more influenced by E2 and leptin than SCN signals [51, 52], the effects of circadian dysregulation (e.g., skipping meals, shift work, and transmeridian travel) as a factor affecting infertility cannot be overlooked [53-55] (Figure 2).
2.2. Ovarian circadian clock
In the rat ovary, clock genes associated with the ovulation cycle have been identified. The day of proestrus saw a considerable increase in BMAL1 expression following the LH surge [15]. In follicular development, Per1 and Per2 mRNA were localized to steroidogenic cells in preantral, antral, and preovulatory follicles, corpora lutea, and interstitial glandular tissue by in situ hybridization histochemistry [56]. Furthermore, Per1 and Per2 mRNA and proteins oscillate in a circadian manner in follicles, granulosa cells, and theca cells. In contrast, LH promotes Per1 as well as Bmal1 expression in the ovary [15]. These clock genes display different amplitudes at different stages of estrus cycle, suggesting endocrine control of the circadian clock [16]. Importantly, gonadotropins also control the ovarian clock, which is supported by experiments indicating that the administration of gonadotropins can synchronize isolated ovaries [57].Using a Per1-luciferase reporter assay, circadian rhythms were noted in the ovaries, and clock gene phasing was observed in response to LH and FSH [57]. In addition, the ovaries of a mouse model of polycystic ovarian syndrome (PCOS) were shown to have an anomaly for the time of Per2 rhythm [58].
2.3. Endometrial circadian clock
Analysis of the relationship between the decidual circadian rhythm and recurrent miscarriage showed that BMAL1 expression in the human decidua during early pregnancy was decreased in patients that experienced recurrent miscarriage [59]. In particular, knockdown may impair the regulation of trophoblast invasion by decidual cells, disrupting proper placenta formation. Polymorphisms in the circadian clock genes are also associated with a higher risk of miscarriage and gene variants were found in BMAL1 and NPAS2. [60]. Furthermore, progesterone is known to affect the peripheral endometrial clock rhythm in humans. When progesterone acts on the endometrium and decidualization occurs, the level of PER1 in the endometrium increases [61]. The microenvironment of the uterus responds to circadian rhythms and adapts to physiological functions. During pregnancy, the fetus is continuously exposed to hormonal and nutritional signals in the maternal endometrium [62, 63]. These results suggest that circadian rhythms play significant roles in reproduction.”
Line 82-85: How come the authors can conclude as such with those pieces of independent, unlinked information?
Thank you. As the reviewer suggested, we have deleted this conclusive sentence.
The title “Circadian clock regulation of gonadotropin” and contents doesn’t quite match. This title may give the impression of circadian clock directly regulate gonadotropin production and secretion. I suggest authors to replace the word of “gonadotropin” with “HPG axis” or else. Also, the information about the circadian clock system in Kisspeptin neurons, GnRH neurons, or HPG axis are too small. It is required to be enriched. I also recommend adding GPR54.
Thank you. We replaced the word of “gonadotropin” with “HPG axis”. The title has been changed to “2.1. Circadian clock regulation of HPG axis”. Additionally, we have incorporated GPR 54 in the following manner. “For the GPR54 receptor, kisspeptin serves as the endogenous agonist. It has been found that GPR54 expression can become rhythmic when E2 levels are raised, a behavior that appears to be controlled by intracellular ERβ receptors [64].”
Line 120-123: First sentence sounds like suggesting Bmal1 downregulation may be involved in recurrent miscarriage (no citation). However, second sentence sounds like suppression of Bmal1 may help to attenuate decidualization. It sounds inconsistent for me.
We deleted the second sentence and left only the first sentence to make it easier to read and included a citation. “Analysis of the relationship between the decidual circadian rhythm and recurrent miscarriage showed that BMAL1 expression in the human decidua during early pregnancy was decreased in patients that experienced recurrent miscarriage [59].”
Line 132: It is insufficient, or hard to understand, to conclude as the authors mentioned.
According to the reviewer’s suggestion, we have deleted this conclusive sentence in line 132.
Section 3
In this section, especially the second paragraph, it is suggestive, but not conclusive information to explain the correlation between skipping breakfast and the circadian clock system. If this is the review for “the correlation between nutrition and reproduction”, this is well written. However, the authors put emphasis on the circadian clock system as a link between nutrition and reproduction, the information written in this section seems insufficient. What exactly occur when breakfast is skipped in the circadian clock system and reproduction system (other than dysmenorrhea)? What is the core of the connection? This paragraph begins with the authors’ hypothesis and ends with a proposal. It would be great if the authors add one or two sentence(s) to conclude this paragraph.
Thank you. We have included our recently published scientific paper in this section.
“Feeding rhythms are important for animals because food entrainable oscillators are located within peripheral tissues and these peripheral oscillators are independent of the SCN [65-67]. We have found that time-restricted feeding regulates a circadian rhythm of the uterine clock that is synchronized throughout the uterine body[68].”
Line 207-214: Here the authors explain the effects of skipping breakfast or longer duration of fasting in circadian clocks. However, nutritional signals mostly influence circadian clock systems in peripheral metabolic organs (e.g., liver, muscle, etc.). Is there any reasonable evidence to link skipping breakfast with reproduction through the circadian clock system? For example, circadian clock expression patterns in ovary or uterus, Kisspeptin circadian expression patterns, etc.
Thank you. This is an important point. We have recently published that circadian clock expression pattern in uterus is controlled by nutritional signals. We have included our findings in our manuscript. “We have found that time-restricted feeding regulates a circadian rhythm of the uterine clock that is synchronized throughout the uterine body[68].”
Section 4
The subject of this section is very interesting. However, as same as section 3, it is too weak to link sexual maturation between circadian clocks with current style. Do EDCs influence in circadian oscillation? Is there any EDCs as ligands directly regulate the circadian clock system (e.g., RORs/REV-ERBs ligands)? Is there any evidence to reveal EDCs influence circadian clock system and its downstream reproduction system (e.g., Kisspeptin)?
Thank you. It was reported that 4-hydroxy-2,3,3',4',5-pentachlorobiphenyl showed altered expression of clock genes including BMAL1. According to the reviewer’s comments, we have included sentences as follows. “The liver of adult male Wistar rats treated with 4-hydroxy-2,3,3',4',5-pentachlorobiphenyl showed altered expression of clock genes including BMAL1 [69].”
Section 5
Line 224: I suggest the authors to say more positive way rather than saying “We hypothesize that”. The hypothesis is a hypothesis. “We expect ~/We hope for ~” would sound better.
Thank you for this constructive comment.
Minor concerns
Line 35-37: “core of molecular interaction in the circadian rhythm” sounds inaccurate. It is true that the BMAL1/CLOCK and PERs/CRYs interact each other to regulate circadian gene expression patterns as a core mechanism of the circadian clock system. What exactly authors want to explain with the words of “core of molecular interaction”?
Thank you. We revised this sentence as follows in line 68-69. “The transcription-translation feedback loop involving CLOCK-BMAL1 and PER heterodimerizes with CRY interact each other to regulate circadian gene expression patterns [30, 31].”
Line 38-39: The paper authors referred is the paper for circadian clock system in bone metabolism. Is that paper the most relevant?
Thank you for pointing this out. We have replaced references in this sentence as follows. “Significant alterations in the circadian behavioral rhythms have been seen in Bmal1-knockout mice, Clock mutant mice, and Per- and Cry-deficient animals [32-35].”
Line 42: “the core SCN and the shell SCN.” > “the core and shell SCN.”
Line 43 and 44: “cellular bodies” > “cell bodies”
Line 47-49: How important? No explanation of that report? How about males?
Line 50-61: This part is one of the cores of this manuscript. I strongly suggest authors to reorganize this part, especially enrich the human part, and bring to the first section of the introduction. Especially, even there are only “few studies” conducted, it is very important to indicate the evidence of reproductive concerns which possibly link to circadian rhythms.
Line 62-71: Again, this part contains some important information related to human physiology and pathology. I recommend authors to marge this section with Line 59-61 and bring to the opening sentence.
Line 94-95: It seems inconsistent with the referred paper.
Line 97-100: Is “dysfunction” the word exactly the authors want to say? “dysregulation”? or “dysfunction caused by ….”? Please check it.
Line 108: “local peripheral circadian regulation” of what? Follicular development itself? Does that event such dynamic?
Line 109: “in a circadian rhythm” > “in a circadian manner”
Line 111-112: “These clock genes have different amplitudes that depend on the estrous cycle” > “These clock genes display (show) different amplitudes (and rhythmic patterns?) at different stages of estrus cycle”
Line 112-114: I suggest the authors to merge line 112-113 and 113-114 as one sentence, remove unnecessary part and, if possible, add a bit more information about the study in Ref#57.
Line 114-116: How abnormal?
Line 116: “Per1 luciferase reporter assay” supposed to be “Per1-luc (or Per1-luciferase) reporter assay”. With or without “-”, the meaning is totally different.
Line 125: What exactly means in “genes that control circadian rhythms”? If the authors want to say, “circadian clock genes” or “core clock genes”, it is a bit circumlocutory.
Line 134-135: It is already mentioned the circadian clock expressions above. Is there any specific reason not to mention here?
Line 144: “conditional knock out” > “conditional knockout”
Line 173: “Time-limited feeding (TRF)” supposed to be “Time-restricted feeding (TRF)”
We appreciate the reviewer's detailed check. Thank you very much for making this review easier to read for our readers. We believe that this manuscript addresses the interests of the broad readership of the International Journal of Molecular Sciences.
References
- Mahoney, M. M., Shift work, jet lag, and female reproduction. Int J Endocrinol 2010, 2010, 813764.
- Summa, K. C.; Vitaterna, M. H.; Turek, F. W., Environmental perturbation of the circadian clock disrupts pregnancy in the mouse. PLoS One 2012, 7, (5), e37668.
- Miller, B. H.; Olson, S. L.; Turek, F. W.; Levine, J. E.; Horton, T. H.; Takahashi, J. S., Circadian clock mutation disrupts estrous cyclicity and maintenance of pregnancy. Curr Biol 2004, 14, (15), 1367-73.
- Bittman, E. L., Circadian Function in Multiple Cell Types Is Necessary for Proper Timing of the Preovulatory LH Surge. J Biol Rhythms 2019, 34, (6), 622-633.
- Boden, M. J.; Varcoe, T. J.; Voultsios, A.; Kennaway, D. J., Reproductive biology of female Bmal1 null mice. Reproduction 2010, 139, (6), 1077-90.
- Sellix, M. T., Clocks underneath: the role of peripheral clocks in the timing of female reproductive physiology. Front Endocrinol (Lausanne) 2013, 4, 91.
- Mereness, A. L.; Murphy, Z. C.; Forrestel, A. C.; Butler, S.; Ko, C.; Richards, J. S.; Sellix, M. T., Conditional Deletion of Bmal1 in Ovarian Theca Cells Disrupts Ovulation in Female Mice. Endocrinology 2016, 157, (2), 913-27.
- Xu, J.; Li, Y.; Wang, Y.; Xu, Y.; Zhou, C., Loss of Bmal1 decreases oocyte fertilization, early embryo development and implantation potential in female mice. Zygote 2016, 24, (5), 760-7.
- Perez, S.; Murias, L.; Fernandez-Plaza, C.; Diaz, I.; Gonzalez, C.; Otero, J.; Diaz, E., Evidence for clock genes circadian rhythms in human full-term placenta. Syst Biol Reprod Med 2015, 61, (6), 360-6.
- Muter, J.; Lucas, E. S.; Chan, Y. W.; Brighton, P. J.; Moore, J. D.; Lacey, L.; Quenby, S.; Lam, E. W.; Brosens, J. J., The clock protein period 2 synchronizes mitotic expansion and decidual transformation of human endometrial stromal cells. FASEB J 2015, 29, (4), 1603-14.
- Ratajczak, C. K.; Herzog, E. D.; Muglia, L. J., Clock gene expression in gravid uterus and extra-embryonic tissues during late gestation in the mouse. Reprod Fertil Dev 2010, 22, (5), 743-50.
- Kennaway, D. J.; Boden, M. J.; Varcoe, T. J., Circadian rhythms and fertility. Mol Cell Endocrinol 2012, 349, (1), 56-61.
- Sen, A.; Hoffmann, H. M., Role of core circadian clock genes in hormone release and target tissue sensitivity in the reproductive axis. Mol Cell Endocrinol 2020, 501, 110655.
- Pan, X.; Taylor, M. J.; Cohen, E.; Hanna, N.; Mota, S., Circadian Clock, Time-Restricted Feeding and Reproduction. Int J Mol Sci 2020, 21, (3), 831.
- Karman, B. N.; Tischkau, S. A., Circadian clock gene expression in the ovary: Effects of luteinizing hormone. Biol Reprod 2006, 75, (4), 624-32.
- Nakamura, T. J.; Sellix, M. T.; Kudo, T.; Nakao, N.; Yoshimura, T.; Ebihara, S.; Colwell, C. S.; Block, G. D., Influence of the estrous cycle on clock gene expression in reproductive tissues: effects of fluctuating ovarian steroid hormone levels. Steroids 2010, 75, (3), 203-12.
- Yaw, A. M.; Duong, T. V.; Nguyen, D.; Hoffmann, H. M., Circadian rhythms in the mouse reproductive axis during the estrous cycle and pregnancy. J Neurosci Res 2021, 99, (1), 294-308.
- Mieda, M.; Okamoto, H.; Sakurai, T., Manipulating the Cellular Circadian Period of Arginine Vasopressin Neurons Alters the Behavioral Circadian Period. Curr Biol 2016, 26, (18), 2535-2542.
- Sun, W.; Li, S. X.; Wang, G.; Dong, S.; Jiang, Y.; Spruyt, K.; Ling, J.; Zhu, Q.; Lee, T. M.; Jiang, F., Association of Sleep and Circadian Activity Rhythm with Emotional Face Processing among 12-month-old Infants. Sci Rep 2018, 8, (1), 3200.
- Ikeda, Y.; Kumagai, H.; Skach, A.; Sato, M.; Yanagisawa, M., Modulation of circadian glucocorticoid oscillation via adrenal opioid-CXCR7 signaling alters emotional behavior. Cell 2013, 155, (6), 1323-36.
- Page, A. J.; Christie, S.; Symonds, E.; Li, H., Circadian regulation of appetite and time restricted feeding. Physiol Behav 2020, 220, 112873.
- Scheer, F. A.; Morris, C. J.; Shea, S. A., The internal circadian clock increases hunger and appetite in the evening independent of food intake and other behaviors. Obesity (Silver Spring) 2013, 21, (3), 421-3.
- Baker, F. C.; Driver, H. S., Circadian rhythms, sleep, and the menstrual cycle. Sleep Med 2007, 8, (6), 613-22.
- Lawson, C. C.; Whelan, E. A.; Lividoti Hibert, E. N.; Spiegelman, D.; Schernhammer, E. S.; Rich-Edwards, J. W., Rotating shift work and menstrual cycle characteristics. Epidemiology 2011, 22, (3), 305-12.
- Stocker, L. J.; Macklon, N. S.; Cheong, Y. C.; Bewley, S. J., Influence of shift work on early reproductive outcomes: a systematic review and meta-analysis. Obstet Gynecol 2014, 124, (1), 99-110.
- Bonde, J. P.; Jorgensen, K. T.; Bonzini, M.; Palmer, K. T., Miscarriage and occupational activity: a systematic review and meta-analysis regarding shift work, working hours, lifting, standing, and physical workload. Scand J Work Environ Health 2013, 39, (4), 325-34.
- Grajewski, B.; Whelan, E. A.; Lawson, C. C.; Hein, M. J.; Waters, M. A.; Anderson, J. L.; MacDonald, L. A.; Mertens, C. J.; Tseng, C. Y.; Cassinelli, R. T., 2nd; Luo, L., Miscarriage among flight attendants. Epidemiology 2015, 26, (2), 192-203.
- Suzumori, N.; Ebara, T.; Matsuki, T.; Yamada, Y.; Kato, S.; Omori, T.; Saitoh, S.; Kamijima, M.; Sugiura-Ogasawara, M.; Japan, E.; Children's Study, G., Effects of long working hours and shift work during pregnancy on obstetric and perinatal outcomes: A large prospective cohort study-Japan Environment and Children's Study. Birth 2020, 47, (1), 67-79.
- Patil, D.; Enquobahrie, D. A.; Peckham, T.; Seixas, N.; Hajat, A., Retrospective cohort study of the association between maternal employment precarity and infant low birth weight in women in the USA. BMJ Open 2020, 10, (1), e029584.
- Xie, Y.; Tang, Q.; Chen, G.; Xie, M.; Yu, S.; Zhao, J.; Chen, L., New Insights Into the Circadian Rhythm and Its Related Diseases. Front Physiol 2019, 10, 682.
- Rijo-Ferreira, F.; Takahashi, J. S., Genomics of circadian rhythms in health and disease. Genome Med 2019, 11, (1), 82.
- Abbas, S.; Ahmed, I.; Kudo, T.; Iqbal, M.; Lee, Y. J.; Fujiwara, T.; Ohkuma, M., A heavy metal tolerant novel bacterium, Bacillus malikii sp. nov., isolated from tannery effluent wastewater. Antonie Van Leeuwenhoek 2015, 108, (6), 1319-1330.
- Miller, B. H.; Olson, S. L.; Levine, J. E.; Turek, F. W.; Horton, T. H.; Takahashi, J. S., Vasopressin regulation of the proestrous luteinizing hormone surge in wild-type and Clock mutant mice. Biol Reprod 2006, 75, (5), 778-84.
- Pendergast, J. S.; Oda, G. A.; Niswender, K. D.; Yamazaki, S., Period determination in the food-entrainable and methamphetamine-sensitive circadian oscillator(s). Proc Natl Acad Sci U S A 2012, 109, (35), 14218-23.
- De Bundel, D.; Gangarossa, G.; Biever, A.; Bonnefont, X.; Valjent, E., Cognitive dysfunction, elevated anxiety, and reduced cocaine response in circadian clock-deficient cryptochrome knockout mice. Front Behav Neurosci 2013, 7, 152.
- Mohawk, J. A.; Green, C. B.; Takahashi, J. S., Central and peripheral circadian clocks in mammals. Annu Rev Neurosci 2012, 35, 445-62.
- Whitmore, D.; Foulkes, N. S.; Sassone-Corsi, P., Light acts directly on organs and cells in culture to set the vertebrate circadian clock. Nature 2000, 404, (6773), 87-91.
- Bass, J., Circadian topology of metabolism. Nature 2012, 491, (7424), 348-56.
- Moore, R. Y.; Speh, J. C.; Leak, R. K., Suprachiasmatic nucleus organization. Cell Tissue Res 2002, 309, (1), 89-98.
- Putteeraj, M.; Soga, T.; Ubuka, T.; Parhar, I. S., A "Timed" Kiss Is Essential for Reproduction: Lessons from Mammalian Studies. Front Endocrinol (Lausanne) 2016, 7, 121.
- Robertson, J. L.; Clifton, D. K.; de la Iglesia, H. O.; Steiner, R. A.; Kauffman, A. S., Circadian regulation of Kiss1 neurons: implications for timing the preovulatory gonadotropin-releasing hormone/luteinizing hormone surge. Endocrinology 2009, 150, (8), 3664-71.
- Comninos, A. N.; Wall, M. B.; Demetriou, L.; Shah, A. J.; Clarke, S. A.; Narayanaswamy, S.; Nesbitt, A.; Izzi-Engbeaya, C.; Prague, J. K.; Abbara, A.; Ratnasabapathy, R.; Salem, V.; Nijher, G. M.; Jayasena, C. N.; Tanner, M.; Bassett, P.; Mehta, A.; Rabiner, E. A.; Honigsperger, C.; Silva, M. R.; Brandtzaeg, O. K.; Lundanes, E.; Wilson, S. R.; Brown, R. C.; Thomas, S. A.; Bloom, S. R.; Dhillo, W. S., Kisspeptin modulates sexual and emotional brain processing in humans. J Clin Invest 2017, 127, (2), 709-719.
- Dror, T.; Franks, J.; Kauffman, A. S., Analysis of multiple positive feedback paradigms demonstrates a complete absence of LH surges and GnRH activation in mice lacking kisspeptin signaling. Biol Reprod 2013, 88, (6), 146.
- Saedi, S.; Khoradmehr, A.; Mohammad Reza, J. S.; Tamadon, A., The role of neuropeptides and neurotransmitters on kisspeptin/kiss1r-signaling in female reproduction. J Chem Neuroanat 2018, 92, 71-82.
- Kunimura, Y.; Iwata, K.; Ishigami, A.; Ozawa, H., Age-related alterations in hypothalamic kisspeptin, neurokinin B, and dynorphin neurons and in pulsatile LH release in female and male rats. Neurobiol Aging 2017, 50, 30-38.
- Garcia, J. P.; Guerriero, K. A.; Keen, K. L.; Kenealy, B. P.; Seminara, S. B.; Terasawa, E., Kisspeptin and Neurokinin B Signaling Network Underlies the Pubertal Increase in GnRH Release in Female Rhesus Monkeys. Endocrinology 2017, 158, (10), 3269-3280.
- Qiu, J.; Nestor, C. C.; Zhang, C.; Padilla, S. L.; Palmiter, R. D.; Kelly, M. J.; Ronnekleiv, O. K., High-frequency stimulation-induced peptide release synchronizes arcuate kisspeptin neurons and excites GnRH neurons. Elife 2016, 5.
- Kalil, B.; Ribeiro, A. B.; Leite, C. M.; Uchoa, E. T.; Carolino, R. O.; Cardoso, T. S.; Elias, L. L.; Rodrigues, J. A.; Plant, T. M.; Poletini, M. O.; Anselmo-Franci, J. A., The Increase in Signaling by Kisspeptin Neurons in the Preoptic Area and Associated Changes in Clock Gene Expression That Trigger the LH Surge in Female Rats Are Dependent on the Facilitatory Action of a Noradrenaline Input. Endocrinology 2016, 157, (1), 323-35.
- Adams, C.; Stroberg, W.; DeFazio, R. A.; Schnell, S.; Moenter, S. M., Gonadotropin-Releasing Hormone (GnRH) Neuron Excitability Is Regulated by Estradiol Feedback and Kisspeptin. J Neurosci 2018, 38, (5), 1249-1263.
- Williams, W. P., 3rd; Jarjisian, S. G.; Mikkelsen, J. D.; Kriegsfeld, L. J., Circadian control of kisspeptin and a gated GnRH response mediate the preovulatory luteinizing hormone surge. Endocrinology 2011, 152, (2), 595-606.
- Smith, J. T.; Acohido, B. V.; Clifton, D. K.; Steiner, R. A., KiSS-1 neurones are direct targets for leptin in the ob/ob mouse. J Neuroendocrinol 2006, 18, (4), 298-303.
- Navarro, V. M.; Castellano, J. M.; McConkey, S. M.; Pineda, R.; Ruiz-Pino, F.; Pinilla, L.; Clifton, D. K.; Tena-Sempere, M.; Steiner, R. A., Interactions between kisspeptin and neurokinin B in the control of GnRH secretion in the female rat. Am J Physiol Endocrinol Metab 2011, 300, (1), E202-10.
- Laposky, A. D.; Bradley, M. A.; Williams, D. L.; Bass, J.; Turek, F. W., Sleep-wake regulation is altered in leptin-resistant (db/db) genetically obese and diabetic mice. Am J Physiol Regul Integr Comp Physiol 2008, 295, (6), R2059-66.
- Sutton, G. M.; Centanni, A. V.; Butler, A. A., Protein malnutrition during pregnancy in C57BL/6J mice results in offspring with altered circadian physiology before obesity. Endocrinology 2010, 151, (4), 1570-80.
- Ando, H.; Kumazaki, M.; Motosugi, Y.; Ushijima, K.; Maekawa, T.; Ishikawa, E.; Fujimura, A., Impairment of peripheral circadian clocks precedes metabolic abnormalities in ob/ob mice. Endocrinology 2011, 152, (4), 1347-54.
- Fahrenkrug, J.; Georg, B.; Hannibal, J.; Hindersson, P.; Gras, S., Diurnal rhythmicity of the clock genes Per1 and Per2 in the rat ovary. Endocrinology 2006, 147, (8), 3769-76.
- Yoshikawa, T.; Sellix, M.; Pezuk, P.; Menaker, M., Timing of the ovarian circadian clock is regulated by gonadotropins. Endocrinology 2009, 150, (9), 4338-47.
- Mereness, A. L.; Murphy, Z. C.; Sellix, M. T., Developmental programming by androgen affects the circadian timing system in female mice. Biol Reprod 2015, 92, (4), 88.
- Lv, S.; Wang, N.; Ma, J.; Li, W. P.; Chen, Z. J.; Zhang, C., Impaired decidualization caused by downregulation of circadian clock gene BMAL1 contributes to human recurrent miscarriagedagger. Biol Reprod 2019, 101, (1), 138-147.
- Kovanen, L.; Saarikoski, S. T.; Aromaa, A.; Lonnqvist, J.; Partonen, T., ARNTL (BMAL1) and NPAS2 gene variants contribute to fertility and seasonality. PLoS One 2010, 5, (4), e10007.
- Zhang, Y.; Meng, N.; Bao, H.; Jiang, Y.; Yang, N.; Wu, K.; Wu, J.; Wang, H.; Kong, S.; Zhang, Y., Circadian gene PER1 senses progesterone signal during human endometrial decidualization. J Endocrinol 2019.
- Seron-Ferre, M.; Valenzuela, G. J.; Torres-Farfan, C., Circadian clocks during embryonic and fetal development. Birth Defects Res C Embryo Today 2007, 81, (3), 204-14.
- Akiyama, S.; Ohta, H.; Watanabe, S.; Moriya, T.; Hariu, A.; Nakahata, N.; Chisaka, H.; Matsuda, T.; Kimura, Y.; Tsuchiya, S.; Tei, H.; Okamura, K.; Yaegashi, N., The uterus sustains stable biological clock during pregnancy. Tohoku J Exp Med 2010, 221, (4), 287-98.
- Tonsfeldt, K. J.; Goodall, C. P.; Latham, K. L.; Chappell, P. E., Oestrogen induces rhythmic expression of the Kisspeptin-1 receptor GPR54 in hypothalamic gonadotrophin-releasing hormone-secreting GT1-7 cells. J Neuroendocrinol 2011, 23, (9), 823-30.
- Stephan, F. K., Phase shifts of circadian rhythms in activity entrained to food access. Physiol Behav 1984, 32, (4), 663-71.
- Mistlberger, R. E., Circadian food-anticipatory activity: formal models and physiological mechanisms. Neurosci Biobehav Rev 1994, 18, (2), 171-95.
- Vollmers, C.; Gill, S.; DiTacchio, L.; Pulivarthy, S. R.; Le, H. D.; Panda, S., Time of feeding and the intrinsic circadian clock drive rhythms in hepatic gene expression. Proc Natl Acad Sci U S A 2009, 106, (50), 21453-8.
- Hosono, T.; Ono, M.; Daikoku, T.; Mieda, M.; Nomura, S.; Kagami, K.; Iizuka, T.; Nakata, R.; Fujiwara, T.; Fujiwara, H.; Ando, H., Time-Restricted Feeding Regulates Circadian Rhythm of Murine Uterine Clock. Curr Dev Nutr 2021, 5, (5), nzab064.
- Ochiai, M.; Iida, M.; Agusa, T.; Takaguchi, K.; Fujii, S.; Nomiyama, K.; Iwata, H., Effects of 4-Hydroxy-2,3,3',4',5-Pentachlorobiphenyl (4-OH-CB107) on Liver Transcriptome in Rats: Implication in the Disruption of Circadian Rhythm and Fatty Acid Metabolism. Toxicol Sci 2018, 165, (1), 118-130.

Reviewer 2 Report
The circadian rhythm, which is necessary for mammalian or avian reproduction, is the medium for environmental and animal reproductive activities. Various tissues and life activities of animals are regulated by circadian clock genes. In this manuscript, the authors started their research on the role of clock genes in female mouse pregnancy and then extend to the regulation of the circadian clock in the whole animal reproductive relative tissues, even involved in nutrition. The content is substantial and clear. It is a good review article and can provide help for further study. But some parts need to be improved and added.
1. Puberty is a key stage for mammalian sexual maturation and reproduction. it also received circadian rhythm regulation, and please review relevant studies.
2. In animals (especially in mice, rats, sheep, and hamsters), the TSH secretion in the PT (pars tuberalis) is very important for the rhythm regulation of reproduction. please review relevant studies.
Author Response
Manuscript #ijms-2006312R1
Title: The Circadian Clock, Nutritional Signals and Reproduction: A Close Relationship
Reviewer’s Comments to Author:
Reviewer #2
Comments and Suggestions for Authors
The circadian rhythm, which is necessary for mammalian or avian reproduction, is the medium for environmental and animal reproductive activities. Various tissues and life activities of animals are regulated by circadian clock genes. In this manuscript, the authors started their research on the role of clock genes in female mouse pregnancy and then extend to the regulation of the circadian clock in the whole animal reproductive relative tissues, even involved in nutrition. The content is substantial and clear. It is a good review article and can provide help for further study. But some parts need to be improved and added.
Thank for the constructive comments for this work and we sincerely appreciate this opportunity to improve our manuscript.
- Puberty is a key stage for mammalian sexual maturation and reproduction. it also received circadian rhythm regulation, and please review relevant studies.
Thank you. We included section 4 as follows.
“4. The circadian clock and puberty
Proper timing of sexual maturation is necessary for reproduction [1-3]. Circadian regulation of reproductive organs is associated with the timing of GnRH release and gonadotropin secretion, and these processes affect sexual maturation [4, 5]. Moreover, human and animal puberty relies on complex endocrine regulation [6]. In European sea bass, a prolonged photoperiod delays or prevents puberty and the release of hormones associated with reproduction [7, 8]. One variable in female puberty is the age at menarche, and the timing of menarche is impacted by light. In women who are blind with loss of light perception, menarche occurs earlier than in women with normal light perception [9]. In addition, women are more likely to experience precocious puberty than men [10, 11]. From a disease perspective, associations between the timing of puberty and risk of developing endometrial or breast cancer in women and prostate cancer in men have been described [12]. Thus, focusing on circadian rhythms may provide clues to preventing and/or treating these diseases.”
- In animals (especially in mice, rats, sheep, and hamsters), the TSH secretion in the PT (pars tuberalis) is very important for the rhythm regulation of reproduction. please review relevant studies.
Thank you. According to the reviewer’s suggestions, we have included sentences regarding PT (pars tuberalis) in the section “2.4. Animal studies on the circadian clock and reproductive function” as follows.
“The pars tuberalis is situated between the anterior lobe of the pituitary gland and the median eminence. It has been demonstrated that melatonin acts as a photoperiodic signal, synchronizing an endogenous oscillator in the pars tuberalis to the photoperiod [13]. Thyroid-stimulating hormone beta (TSH) cells are found in the pars tuberalis, which also triggers the secretion of TSH. TSH promotes triiodothyronine synthesis, which helps gonadotropin-releasing hormone-I release, luteinizing hormone and follicle stimulating hormone release [14]. Recent research has shown that pars tuberalis controls seasonal reproduction with its TSH secretion [15, 16].”
We greatly appreciate the opportunity to revise our manuscript and to respond to comments and questions from the reviewers and editors.
References
- Simonneaux, V.; Bahougne, T.; Angelopoulou, E., Daily rhythms count for female fertility. Best Pract Res Clin Endocrinol Metab 2017, 31, (5), 505-519.
- Sellix, M. T.; Murphy, Z. C.; Menaker, M., Excess androgen during puberty disrupts circadian organization in female rats. Endocrinology 2013, 154, (4), 1636-47.
- Huhtaniemi, I., Mutations along the pituitary-gonadal axis affecting sexual maturation: novel information from transgenic and knockout mice. Mol Cell Endocrinol 2006, 254-255, 84-90.
- Resuehr, H. E.; Resuehr, D.; Olcese, J., Induction of mPer1 expression by GnRH in pituitary gonadotrope cells involves EGR-1. Mol Cell Endocrinol 2009, 311, (1-2), 120-5.
- Chu, A.; Zhu, L.; Blum, I. D.; Mai, O.; Leliavski, A.; Fahrenkrug, J.; Oster, H.; Boehm, U.; Storch, K. F., Global but not gonadotrope-specific disruption of Bmal1 abolishes the luteinizing hormone surge without affecting ovulation. Endocrinology 2013, 154, (8), 2924-35.
- Perry, J. R.; Day, F.; Elks, C. E.; Sulem, P.; Thompson, D. J.; Ferreira, T.; He, C.; Chasman, D. I.; Esko, T.; Thorleifsson, G.; Albrecht, E.; Ang, W. Q.; Corre, T.; Cousminer, D. L.; Feenstra, B.; Franceschini, N.; Ganna, A.; Johnson, A. D.; Kjellqvist, S.; Lunetta, K. L.; McMahon, G.; Nolte, I. M.; Paternoster, L.; Porcu, E.; Smith, A. V.; Stolk, L.; Teumer, A.; Tsernikova, N.; Tikkanen, E.; Ulivi, S.; Wagner, E. K.; Amin, N.; Bierut, L. J.; Byrne, E. M.; Hottenga, J. J.; Koller, D. L.; Mangino, M.; Pers, T. H.; Yerges-Armstrong, L. M.; Zhao, J. H.; Andrulis, I. L.; Anton-Culver, H.; Atsma, F.; Bandinelli, S.; Beckmann, M. W.; Benitez, J.; Blomqvist, C.; Bojesen, S. E.; Bolla, M. K.; Bonanni, B.; Brauch, H.; Brenner, H.; Buring, J. E.; Chang-Claude, J.; Chanock, S.; Chen, J.; Chenevix-Trench, G.; Collee, J. M.; Couch, F. J.; Couper, D.; Coveillo, A. D.; Cox, A.; Czene, K.; D'Adamo A, P.; Smith, G. D.; De Vivo, I.; Demerath, E. W.; Dennis, J.; Devilee, P.; Dieffenbach, A. K.; Dunning, A. M.; Eiriksdottir, G.; Eriksson, J. G.; Fasching, P. A.; Ferrucci, L.; Flesch-Janys, D.; Flyger, H.; Foroud, T.; Franke, L.; Garcia, M. E.; Garcia-Closas, M.; Geller, F.; de Geus, E. E.; Giles, G. G.; Gudbjartsson, D. F.; Gudnason, V.; Guenel, P.; Guo, S.; Hall, P.; Hamann, U.; Haring, R.; Hartman, C. A.; Heath, A. C.; Hofman, A.; Hooning, M. J.; Hopper, J. L.; Hu, F. B.; Hunter, D. J.; Karasik, D.; Kiel, D. P.; Knight, J. A.; Kosma, V. M.; Kutalik, Z.; Lai, S.; Lambrechts, D.; Lindblom, A.; Magi, R.; Magnusson, P. K.; Mannermaa, A.; Martin, N. G.; Masson, G.; McArdle, P. F.; McArdle, W. L.; Melbye, M.; Michailidou, K.; Mihailov, E.; Milani, L.; Milne, R. L.; Nevanlinna, H.; Neven, P.; Nohr, E. A.; Oldehinkel, A. J.; Oostra, B. A.; Palotie, A.; Peacock, M.; Pedersen, N. L.; Peterlongo, P.; Peto, J.; Pharoah, P. D.; Postma, D. S.; Pouta, A.; Pylkas, K.; Radice, P.; Ring, S.; Rivadeneira, F.; Robino, A.; Rose, L. M.; Rudolph, A.; Salomaa, V.; Sanna, S.; Schlessinger, D.; Schmidt, M. K.; Southey, M. C.; Sovio, U.; Stampfer, M. J.; Stockl, D.; Storniolo, A. M.; Timpson, N. J.; Tyrer, J.; Visser, J. A.; Vollenweider, P.; Volzke, H.; Waeber, G.; Waldenberger, M.; Wallaschofski, H.; Wang, Q.; Willemsen, G.; Winqvist, R.; Wolffenbuttel, B. H.; Wright, M. J.; Australian Ovarian Cancer, S.; Network, G.; kConFab; LifeLines Cohort, S.; InterAct, C.; Early Growth Genetics, C.; Boomsma, D. I.; Econs, M. J.; Khaw, K. T.; Loos, R. J.; McCarthy, M. I.; Montgomery, G. W.; Rice, J. P.; Streeten, E. A.; Thorsteinsdottir, U.; van Duijn, C. M.; Alizadeh, B. Z.; Bergmann, S.; Boerwinkle, E.; Boyd, H. A.; Crisponi, L.; Gasparini, P.; Gieger, C.; Harris, T. B.; Ingelsson, E.; Jarvelin, M. R.; Kraft, P.; Lawlor, D.; Metspalu, A.; Pennell, C. E.; Ridker, P. M.; Snieder, H.; Sorensen, T. I.; Spector, T. D.; Strachan, D. P.; Uitterlinden, A. G.; Wareham, N. J.; Widen, E.; Zygmunt, M.; Murray, A.; Easton, D. F.; Stefansson, K.; Murabito, J. M.; Ong, K. K., Parent-of-origin-specific allelic associations among 106 genomic loci for age at menarche. Nature 2014, 514, (7520), 92-97.
- Bayarri, M. J.; Rodriguez, L.; Zanuy, S.; Madrid, J. A.; Sanchez-Vazquez, F. J.; Kagawa, H.; Okuzawa, K.; Carrillo, M., Effect of photoperiod manipulation on the daily rhythms of melatonin and reproductive hormones in caged European sea bass (Dicentrarchus labrax). Gen Comp Endocrinol 2004, 136, (1), 72-81.
- Bayarri, M. J.; Zanuy, S.; Yilmaz, O.; Carrillo, M., Effects of continuous light on the reproductive system of European sea bass gauged by alterations of circadian variations during their first reproductive cycle. Chronobiol Int 2009, 26, (2), 184-99.
- Flynn-Evans, E. E.; Stevens, R. G.; Tabandeh, H.; Schernhammer, E. S.; Lockley, S. W., Effect of light perception on menarche in blind women. Ophthalmic Epidemiol 2009, 16, (4), 243-8.
- de Vries, L.; Kauschansky, A.; Shohat, M.; Phillip, M., Familial central precocious puberty suggests autosomal dominant inheritance. J Clin Endocrinol Metab 2004, 89, (4), 1794-800.
- Wehkalampi, K.; Widen, E.; Laine, T.; Palotie, A.; Dunkel, L., Patterns of inheritance of constitutional delay of growth and puberty in families of adolescent girls and boys referred to specialist pediatric care. J Clin Endocrinol Metab 2008, 93, (3), 723-8.
- Day, F. R.; Thompson, D. J.; Helgason, H.; Chasman, D. I.; Finucane, H.; Sulem, P.; Ruth, K. S.; Whalen, S.; Sarkar, A. K.; Albrecht, E.; Altmaier, E.; Amini, M.; Barbieri, C. M.; Boutin, T.; Campbell, A.; Demerath, E.; Giri, A.; He, C.; Hottenga, J. J.; Karlsson, R.; Kolcic, I.; Loh, P. R.; Lunetta, K. L.; Mangino, M.; Marco, B.; McMahon, G.; Medland, S. E.; Nolte, I. M.; Noordam, R.; Nutile, T.; Paternoster, L.; Perjakova, N.; Porcu, E.; Rose, L. M.; Schraut, K. E.; Segre, A. V.; Smith, A. V.; Stolk, L.; Teumer, A.; Andrulis, I. L.; Bandinelli, S.; Beckmann, M. W.; Benitez, J.; Bergmann, S.; Bochud, M.; Boerwinkle, E.; Bojesen, S. E.; Bolla, M. K.; Brand, J. S.; Brauch, H.; Brenner, H.; Broer, L.; Bruning, T.; Buring, J. E.; Campbell, H.; Catamo, E.; Chanock, S.; Chenevix-Trench, G.; Corre, T.; Couch, F. J.; Cousminer, D. L.; Cox, A.; Crisponi, L.; Czene, K.; Davey Smith, G.; de Geus, E.; de Mutsert, R.; De Vivo, I.; Dennis, J.; Devilee, P.; Dos-Santos-Silva, I.; Dunning, A. M.; Eriksson, J. G.; Fasching, P. A.; Fernandez-Rhodes, L.; Ferrucci, L.; Flesch-Janys, D.; Franke, L.; Gabrielson, M.; Gandin, I.; Giles, G. G.; Grallert, H.; Gudbjartsson, D. F.; Guenel, P.; Hall, P.; Hallberg, E.; Hamann, U.; Harris, T. B.; Hartman, C. A.; Heiss, G.; Hooning, M. J.; Hopper, J. L.; Hu, F.; Hunter, D. J.; Ikram, M. A.; Im, H. K.; Jarvelin, M. R.; Joshi, P. K.; Karasik, D.; Kellis, M.; Kutalik, Z.; LaChance, G.; Lambrechts, D.; Langenberg, C.; Launer, L. J.; Laven, J. S. E.; Lenarduzzi, S.; Li, J.; Lind, P. A.; Lindstrom, S.; Liu, Y.; Luan, J.; Magi, R.; Mannermaa, A.; Mbarek, H.; McCarthy, M. I.; Meisinger, C.; Meitinger, T.; Menni, C.; Metspalu, A.; Michailidou, K.; Milani, L.; Milne, R. L.; Montgomery, G. W.; Mulligan, A. M.; Nalls, M. A.; Navarro, P.; Nevanlinna, H.; Nyholt, D. R.; Oldehinkel, A. J.; O'Mara, T. A.; Padmanabhan, S.; Palotie, A.; Pedersen, N.; Peters, A.; Peto, J.; Pharoah, P. D. P.; Pouta, A.; Radice, P.; Rahman, I.; Ring, S. M.; Robino, A.; Rosendaal, F. R.; Rudan, I.; Rueedi, R.; Ruggiero, D.; Sala, C. F.; Schmidt, M. K.; Scott, R. A.; Shah, M.; Sorice, R.; Southey, M. C.; Sovio, U.; Stampfer, M.; Steri, M.; Strauch, K.; Tanaka, T.; Tikkanen, E.; Timpson, N. J.; Traglia, M.; Truong, T.; Tyrer, J. P.; Uitterlinden, A. G.; Edwards, D. R. V.; Vitart, V.; Volker, U.; Vollenweider, P.; Wang, Q.; Widen, E.; van Dijk, K. W.; Willemsen, G.; Winqvist, R.; Wolffenbuttel, B. H. R.; Zhao, J. H.; Zoledziewska, M.; Zygmunt, M.; Alizadeh, B. Z.; Boomsma, D. I.; Ciullo, M.; Cucca, F.; Esko, T.; Franceschini, N.; Gieger, C.; Gudnason, V.; Hayward, C.; Kraft, P.; Lawlor, D. A.; Magnusson, P. K. E.; Martin, N. G.; Mook-Kanamori, D. O.; Nohr, E. A.; Polasek, O.; Porteous, D.; Price, A. L.; Ridker, P. M.; Snieder, H.; Spector, T. D.; Stockl, D.; Toniolo, D.; Ulivi, S.; Visser, J. A.; Volzke, H.; Wareham, N. J.; Wilson, J. F.; LifeLines Cohort, S.; InterAct, C.; kConFab, A. I.; Endometrial Cancer Association, C.; Ovarian Cancer Association, C.; consortium, P.; Spurdle, A. B.; Thorsteindottir, U.; Pollard, K. S.; Easton, D. F.; Tung, J. Y.; Chang-Claude, J.; Hinds, D.; Murray, A.; Murabito, J. M.; Stefansson, K.; Ong, K. K.; Perry, J. R. B., Genomic analyses identify hundreds of variants associated with age at menarche and support a role for puberty timing in cancer risk. Nat Genet 2017, 49, (6), 834-841.
- Wagner, G. C.; Johnston, J. D.; Tournier, B. B.; Ebling, F. J.; Hazlerigg, D. G., Melatonin induces gene-specific effects on rhythmic mRNA expression in the pars tuberalis of the Siberian hamster (Phodopus sungorus). Eur J Neurosci 2007, 25, (2), 485-90.
- Kosonsiriluk, S.; Mauro, L. J.; Chaiworakul, V.; Chaiseha, Y.; El Halawani, M. E., Photoreceptive oscillators within neurons of the premammillary nucleus (PMM) and seasonal reproduction in temperate zone birds. Gen Comp Endocrinol 2013, 190, 149-55.
- Yoshimura, T.; Yasuo, S.; Watanabe, M.; Iigo, M.; Yamamura, T.; Hirunagi, K.; Ebihara, S., Light-induced hormone conversion of T4 to T3 regulates photoperiodic response of gonads in birds. Nature 2003, 426, (6963), 178-81.
- Nakao, N.; Ono, H.; Yamamura, T.; Anraku, T.; Takagi, T.; Higashi, K.; Yasuo, S.; Katou, Y.; Kageyama, S.; Uno, Y.; Kasukawa, T.; Iigo, M.; Sharp, P. J.; Iwasawa, A.; Suzuki, Y.; Sugano, S.; Niimi, T.; Mizutani, M.; Namikawa, T.; Ebihara, S.; Ueda, H. R.; Yoshimura, T., Thyrotrophin in the pars tuberalis triggers photoperiodic response. Nature 2008, 452, (7185), 317-22.

Round 2
Reviewer 1 Report
Re: Manuscript #ijms-2006312R1, Title: The Circadian Clock, Nutritional Signals and Reproduction: A Close Relationship
Dear Authors,
Thank you for giving me a chance to review your revised manuscript. I could acknowledge your prompt and enormous effort in your revised manuscript. I could see that the expertise in the field of reproduction and nutrition.
However, there are some critical points need to be improved.
There are some misunderstandings or possibly authors’ weakness of knowledges for circadian biology. For example, line 25, “These Per and Cry proteins interact with Clock/Bmal1 complexes to form heterodimers and suppress Clock and Bmal1 transcription.”. This is untrue, possibly misunderstanding or type. Clock and Bmal1 composes heterodimer to exert its transcriptional activity by binding on e-boxes. Pers (Per1/2/3) and Crys (Cry1/2) compose “complex” with Clock/Bmal1 heterodimer and inactivate its transcriptional activities.
In addition, line 26, “24-hour cycle is produced in the cells by this negative feedback loop, which is typically synchronized within the organ”. First of all, please accept that the meaning of “Circadian”, and, as it means, Circadian clock systems and rhythms are actually not exactly 24hrs. The negative feedback of Pers/Crys is not solo mechanisms of exerting circadian oscillation. There is also a positive feedback loop by RORs and its competitive suppressor REV-ERBs. In this sentence, it may give wrong impression that the circadian clock system is to compose Clock/Bmal1 as positive and Crys/Pers as negative transcription factors. Feedback loops are more dynamic than just negative feedback loop. In general, in the field of circadian biology, we state those negative and positive transcriptional feedback loops as Transcription-translation feedback loop (TTFL).
This is the very beginning of this review manuscript which is newly added as a revised sentence. It may make a negative impression of readers about the authors’ knowledges especially in the circadian biology field, and the reliability of information throughout this manuscript.
Author Response
Manuscript #ijms-2006312R2
Title: The Circadian Clock, Nutritional Signals and Reproduction: A Close Relationship
Comments and Suggestions for Authors:
Reviewer: 1
Thank you for giving me a chance to review your revised manuscript. I could acknowledge your prompt and enormous effort in your revised manuscript. I could see that the expertise in the field of reproduction and nutrition.
Thank for the constructive comments for this work and we sincerely appreciate this opportunity to improve our manuscript.
There are some misunderstandings or possibly authors’ weakness of knowledges for circadian biology. For example, line 25, “These Per and Cry proteins interact with Clock/Bmal1 complexes to form heterodimers and suppress Clock and Bmal1 transcription.”. This is untrue, possibly misunderstanding or type. Clock and Bmal1 composes heterodimer to exert its transcriptional activity by binding on e-boxes. Pers (Per1/2/3) and Crys (Cry1/2) compose “complex” with Clock/Bmal1 heterodimer and inactivate its transcriptional activities.
Thank you for pointing this out. We revised these sentences as follows. “The transcription of core clock genes period (Per) and cryptochrome (Cry) is activated by the heterodimer of the transcription factors circadian locomotor output cycles kaput (Clock) and brain and muscle arnt-like protein-1 (Bmal1). By binding to E-box sequences in the promoters of Per1/2 and Cry1/2 genes, the CLOCK-BMAL1 heterodimer promotes transcription of these genes. Per1/2 and Cry1/2 compose complex with Clock/Bmal1 heterodimer and inactivate its transcriptional activities.” In addition, we revised Introduction, line 69 as follows. “The CLOCK-BMAL1 heterodimer induces transcription of Per1/2 and Cry1/2 genes by binding to E-box (CACGTG/T) regions in their promoters. Together with the Clock/Bmal1 heterodimer, Per1/2 and Cry1/2 form a complex that inhibits the transcriptional activity of CLOCK-BMAL1 [1, 2].”
In addition, line 26, “24-hour cycle is produced in the cells by this negative feedback loop, which is typically synchronized within the organ”. First of all, please accept that the meaning of “Circadian”, and, as it means, Circadian clock systems and rhythms are actually not exactly 24hrs. The negative feedback of Pers/Crys is not solo mechanisms of exerting circadian oscillation. There is also a positive feedback loop by RORs and its competitive suppressor REV-ERBs. In this sentence, it may give wrong impression that the circadian clock system is to compose Clock/Bmal1 as positive and Crys/Pers as negative transcription factors. Feedback loops are more dynamic than just negative feedback loop. In general, in the field of circadian biology, we state those negative and positive transcriptional feedback loops as Transcription-translation feedback loop (TTFL). This is the very beginning of this review manuscript which is newly added as a revised sentence. It may make a negative impression of readers about the authors’ knowledges especially in the circadian biology field, and the reliability of information throughout this manuscript.
According to the reviewer’s suggestion, we have deleted this sentence in line 26.
Thank you very much for making this review easier to read for our readers. We believe that this manuscript addresses the interests of the broad readership of the International Journal of Molecular Sciences.
References
- Xie, Y.; Tang, Q.; Chen, G.; Xie, M.; Yu, S.; Zhao, J.; Chen, L., New Insights Into the Circadian Rhythm and Its Related Diseases. Front Physiol 2019, 10, 682.
- Rijo-Ferreira, F.; Takahashi, J. S., Genomics of circadian rhythms in health and disease. Genome Med 2019, 11, (1), 82.